# CryptoGCN: Fast and Scalable Homomorphically Encrypted Graph Convolutional Network Inference

**Ran Ran**
Lehigh University
rar418@lehigh.edu

**Nuo Xu**
Lehigh University
nux219@lehigh.edu

**Wei Wang**
Microsoft
wewang3@microsoft.com

**Gang Quan**
Florida International University
gaquan@fiu.edu

**Jieming Yin**
Nanjing University of
Posts and Telecommunications
jieming.yin@njupt.edu.cn

**Wujie Wen**
Lehigh University
wuw219@lehigh.edu

## Abstract

Recently cloud-based graph convolutional network (GCN) has demonstrated great success and potential in many privacy-sensitive applications such as personal healthcare and financial systems. Despite its high inference accuracy and performance on the cloud, maintaining data privacy in GCN inference, which is of paramount importance to these practical applications, remains largely unexplored. In this paper, we take an initial attempt towards this and develop *CryptoGCN*–a homomorphic encryption (HE) based GCN inference framework. A key to the success of our approach is to reduce the tremendous computational overhead for HE operations, which can be orders of magnitude higher than its counterparts in the plaintext space. To this end, we develop a solution that can effectively take advantage of the sparsity of matrix operations in GCN inference to significantly reduce the encrypted computational overhead. Specifically, we propose a novel **A**djacency **M**atrix-**A**ware **(AMA)** data formatting method along with the **AMA** assisted patterned sparse matrix partitioning, to exploit the complex graph structure and perform efficient matrix-matrix multiplication in HE computation. In this way, the number of HE operations can be significantly reduced. We also develop a co-optimization framework that can explore the trade-offs among the accuracy, security level, and computational overhead by judicious pruning and polynomial approximation of activation modules in GCNs. Based on the NTU-XVIEW skeleton joint dataset, i.e., the largest dataset evaluated homomorphically by far as we are aware of, our experimental results demonstrate that *CryptoGCN* outperforms state-of-the-art solutions in terms of the latency and number of homomorphic operations, i.e., achieving as much as a $3.10\times$ speedup on latency and reduces the total Homomorphic Operation Count (HOC) by 77.4% with a small accuracy loss of 1-1.5%. Our code is publicly available at `https://github.com/ranran0523/CryptoGCN`.

## 1 Introduction

Graph Convolutional Neural Networks [23] (GCNs) have recently emerged in machine learning and demonstrated superior performance in various privacy-sensitive applications such as human action recognition [35, 38], financial recommendation system [37], and autonomous driving [25]. While it has been increasingly popular to deploy machine learning services on the cloud, the cloud environment raises critical concerns for GCN-based privacy-sensitive services, since graph data usually contain a considerable amount of sensitive information.

36th Conference on Neural Information Processing Systems (NeurIPS 2022).

Recently, Homomorphic Encryption (HE) based private-preserving machine learning (PPML) has emerged to be an effective way to protect the privacy of clients. Using HE, a client can encrypt the data and send it to the cloud. The cloud server directly operates on the encrypted data and sends the encrypted results back to the client, which can then be decrypted and used. As the data are encrypted throughout the entire process when it is out of the client's control, data privacy is greatly enhanced. The challenge, however, is to deal with the tremendously increased computational cost associated with the HE operations (e.g., rotations, multiplications, additions), which is orders of magnitude higher than that in the plaintext space [13, 31, 20].

There exists a flurry of work that aims to alleviate the computation overhead of HE-based PPML [3, 11, 13, 22, 27]. However, these solutions are all focused on the traditional Convolutional Neural Network (CNN) models and are ineffective for GCNs because of a major difference of computation pattern between GCN and CNN. That is, each graph convolution layer in the deep GCN model would involve a unique adjacency matrix multiplication operation to incorporate the graphic information during the convolution. This additional matrix multiplication in GCN can significantly increase the required HE operations. As our profiling result in Fig. 1a demonstrates, for a typical 64-channel-GCN layer with matrix size $25 \times 25$, the matrix multiplication can lead to a nearly $49\times$ increase in terms of homomorphic operation count (HOC) in the worst case. How to deal with the significantly increased HE operations in the GCN inferencing process presents a primary challenge. Furthermore, current HE frameworks, such as the Cheon-Kim-Kim-Song (CKKS) scheme [7] used by this work, are normally built on the so-called Leveled HE (LHE) [2] scheme, which has a limit on the maximum number of concatenated homomorphic multiplication operations. The extra multiplication operations in GCNs increase the multiplication depths of the total computation circuit, leading to the requirement of larger encryption parameters (e.g. increased polynomial degree and primes) to maintain the same security levels. As shown in Fig. 1c, the choice of parameters for HE schemes not only affects the security level of the encryption but also has profound impacts on the latency of HE operations. How to judiciously choose the HE parameter to optimize the security, computational cost, and latency is also critical for effective and efficient GCN inferencing.

One intuitive approach to dealing with the adjacency matrix multiplication operation at each graph convolution layer is to employ the state-of-the-art encrypted matrix multiplication methods (e.g. [8, 19]). However, these solutions can be problematic for deep GCN models because of increasing the total multiplication depths, e.g. by 3 [8] and 6 [19], which leads to a higher polynomial degree for encryption to maintain at least the same security level. From the comparison in Fig. 1b, with no optimization, these results would be inferior to the method in [14], which we used in our paper as the baseline results for comparison. Instead, we take advantage of the sparsity in the adjacency matrix, which is very common, especially for applications based on a class of popular GCNs–Spatial-Temporal GCN (ST-GCN), and can dramatically reduce the number of HE operations.

In this paper, we have made the following contributions. First, we develop an approach that can effectively take advantage of the sparsity of matrix operations in GCN inferencing that can significantly reduce the computational overhead. As shown above, for GCN inferencing, the required matrix-matrix multiplication can lead to significantly increased HE operations. In the meantime, the matrix operation for GCN inferencing exhibits strong sparsity features, which can be exploited to reduce the computational overhead. To this end, we develop a novel GCN data formatting method, i.e., **A**djacency **M**atrix **A**ware **(AMA)** data formatting method to support the associated multi-channel multi-batch convolution and matrix-matrix multiplications, which can exploit the single instruction multiple data (SIMD) structures in HE computation and thus greatly reduce the HE operations. Second, we also study how to better manage the HE computation numbers and levels for GCN inferencing by judiciously pruning and approximation of the activation module in GCN and settings of HE parameters. We develop a co-optimization framework that can help to explore the tradeoffs among security level, inference accuracy, and inference latency. Third, we conduct extensive experiments based on the NTU-XVIEW [34] skeleton joint [36, 15, 4], dataset. Our experimental results show that the AMA data formatting achieves a latency speedup of up to $3.10\times$, and the Activation Pruning achieves as much as $2.29\times$ speedup for latency. To our best knowledge, this is the first work that builds the HE-based PPML pipeline for GCN-based models with HOC decrease by as much as 77.44% compared to previous benchmarks.

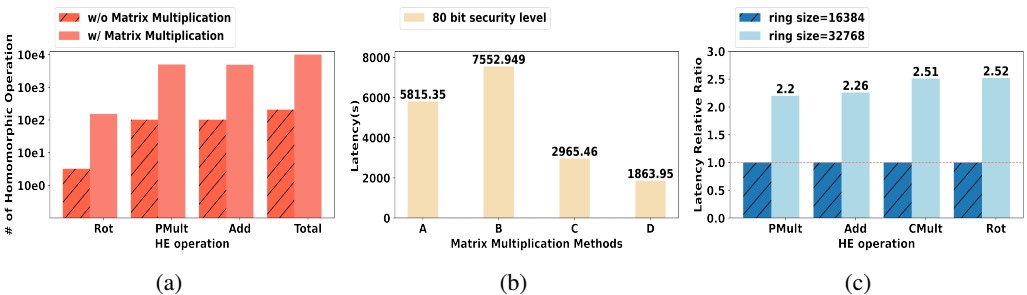

Figure 1: Motivation examples: (a) the number of HE operations increased at log scale (by $\sim 49\times$ in total) due to involving an adjacent matrix-based matrix multiplication in a typical 64-channel GCN layer (detailed setting in Sec. 4); (b) Comparison with the state-of-the-art HE matrix multiplication methods applied in the 64-STGCN-3 model (see Table 2), A [8], B [19], C [14], D-Ours; (c) The latency of HE operations on a single ciphertext increased by up to $2.52\times$ (32K v.s. 16K, normalized to 16K) due to enlarging the polynomial degree. *PMult*, *Add*, *CMult* and *Rot* represent ciphertext-plaintext multiplication, ciphertext addition, ciphertext-ciphertext multiplication and rotation, respectively (see Sec. 2).

## 2  Background and Related Work

**Spatial-Temporal Graph Convolution Network.**  In this work, we focus on a class of popular GCNs–Spatial-Temporal Graph Convolution Network (ST-GCN) [38]. ST-GCN mainly consists of two types of graph convolutions–Spatial convolution and Temporal convolution, which aim to extract Spatial and Temporal information from the input graph data, respectively. The Spatial convolution can be expressed by Equation 1:

$$H = \sum_j \tilde{D}_j^{-\frac{1}{2}} \tilde{A}_j \tilde{D}_j^{-\frac{1}{2}} X W_j \tag{1}$$

Where $j$ indicates different sets of graph connections separated by a graph node partition strategy to better extract the Spatial information. $X$ is the input graph data. $W_j \in \mathbb{R}^{C_{in} \times C_{out}}$ represents a set of filter parameters to transform the input tensor features from $C_{in}$ input channels to $C_{out}$ output channels. $\tilde{D}_j$ is the degree matrix. $\tilde{A}_j$ is the adjacency matrix with self-loop. The $X W_j$ term is implemented by a 2D-convolution with kernel size $1 \times 1$, then multiplied with the normalized adjacency matrix $\tilde{D}_j^{-\frac{1}{2}} \tilde{A}_j \tilde{D}_j^{-\frac{1}{2}}$, and finally the resulted data are summed up as output feature $H$. The Temporal convolution is a convolution operation performed on the same graph node data from different time slices.

**CKKS Leveled Homomorphic Encryption Scheme.**  The Cheon-Kim-Kim-Song (CKKS) scheme [7], which is based on the hardness of ring learning with errors (RLWE) problem, is a leveled homomorphic encryption scheme that allows arithmetic operations on encrypted data over fixed-point numbers. CKKS provides configurable precision by taking the encryption noise [24] as natural error introduced in the approximation computations and through dropping the least significant bits of computations via the rescaling of ciphertext. The supported homomorphic operations include the ciphertext addition *Add(ct$_1$, ct$_2$)*, ciphertext multiplication *CMult(ct$_1$, ct$_2$)*, scalar multiplication *PMult(ct , pt)*, rotation *Rot(ct , k)* and rescaling *Rescale(ct)*. The scalar multiplication is to multiply a ciphertext with plaintext. The rotation is to apply Galois automorphisms of the cyclotomic extension to the plaintext polynomials in encrypted form resulting in a cyclic shift of the slot vector. For instance, $Rot(ct, k)$ transforms an encryption of $(v_0, ..., v_{N/2-1})$ into an encryption of $(v_k, ..., v_{N/2-1}, v_0, ..., v_{k-1})$.

Compared with unencrypted computations, CKKS introduces significant runtime and memory overhead. The use of packing, also referred to as batching, allows to pack of multiple data values into one ciphertext so the encrypted computations can be done in a Single Instruction Multiple Data (SIMD) manner. We use this property to improve the amortized overhead in this paper.

The security level [5] of the CKKS scheme is measured in bits. With $\lambda = 128$, it will take around $2^{128}$ operations to break the encryption. Throughout this paper, we assume that $N$ is the cyclotomic

polynomial degree. CKKS is a leveled HE and the level of a ciphertext ($l$) is defined as the number of successive multiplications that can be performed to the ciphertext without bootstrapping. The level is decreased by one through the rescaling operation after each homomorphic multiplication. If the level becomes zero, bootstrapping is needed to make this zero-level ciphertext a higher-level ciphertext to enable further homomorphic operations. In our work, we optimize the GCN network to have lower depth and select proper parameters to avoid the costly bootstrapping procedure.

**Related Work.** CryptoNets [13] is the first work that demonstrates the feasibility of building PPML by HE. CryproNets evaluates a 5-layer neural network with MNIST dataset and achieves 59000 image inferences within one hour. However, the long inference latency makes it hard to be applied to large-scale models and datasets. A recent work named SHE [9], translates the nonlinear ReLU and Max Pooling operations as Boolean operations to realize the TFHE-based PPML, and achieves the state-of-the-art inference accuracy. However, SHE also incurs a high inference latency, where making a prediction on a CIFAR-10 image costs 2258 seconds. There also exist many MPC solutions which combine the two-party computation protocols [39] with HE frameworks to achieve the low inference latency [32, 21, 29, 16, 26, 30]. However, they suffer from high communication overhead incurred by data transfer. For example, DeepSecure [32] needs to exchange 722GB of data between the client and the server for only a 5-layer CNN inference on one MNIST image. Another approach [10] employs the sparsity location vector to avoid unnecessary computations when performing the secure sparse matrix multiplication. They exploit a PIR protocol [1] to extract the required non-zero elements from the ciphertexts in their multi-party computation (MPC) setting, which is not applicable in our HE environment since decryption at the inference stage is not possible. Other studies such as LoLa [3], CHET [11], and HEAR [22] aim to use the ciphertext packing technique to place multiple values from network nodes in the same ciphertext, so that HE operations can be conducted efficiently via single instruction multiple data (SIMD). However, their data formats for ciphertext are not optimized for GCN-based models, resulting in a large amount of HE operations (multiplication and addition).

## 3 Methodology

In this section, we first present the threat model assumption for this work. We then discuss technique details of our proposed *CryptoGCN*–a CKKS-based homomorphic encryption framework tailored for fast and scalable GCN inference on encrypted graph tensor. In particular, we focus on two key components to significantly reduce the number of HE operations: 1) adjacency matrix-aware (AMA) data formatting dedicated to simplifying GCN's matrix-matrix multiplications without involving ciphertext rotation; 2) non-linear Activation Pruning to reduce the multiplicative depth, resulting in trade-offs between security level and inference latency with low accuracy loss.

**Threat Model.** We assume the cloud-based machine learning service, of which a well-trained graph convolutional network model with plaintext weights, is hosted in a cloud server. A client can upload private and sensitive data to the cloud for obtaining the online inference service. The cloud server is semi-honest (e.g. honest but curious). To ensure the confidentiality of clients' data against such a cloud server, a client will encrypt the data by HE and send it to the cloud for performing encrypted inference without decrypting the data or accessing the client's private key. The client then can decrypt the returned encrypted inference outcome from the cloud using the private key. In this work, we focus on encrypting graph node features, and the normalized adjacency matrix (often sparse and the same for different graph inputs) is assumed as plaintext.

### 3.1 AMA Data Formatting and Matrix-Matrix Multiplication

Since HE operations can be performed on encrypted vectors by taking advantage of the SIMD architecture for parallel computing, the input tensor should be well placed into a "big vector container" by a certain format which we call "data formatting". The state-of-the-art row-major format [11, 22] concatenates a ciphertext row by row and facilitates the dot-product computation that is essential to convolution or matrix-based operations. However, as we shall show later, it cannot efficiently support GCNs' matrix-matrix multiplication that would occur multiple times in a multi-channel GCN layer, because of the extensive ciphertext rotation, addition, and multiplication. The mini-batch inference over multiple GCN layers would further escalate HE overhead and latency for the row-major format.

**Adjacency Matrix-Aware (AMA) Data Formatting.** We use the skeleton-based action recognition graph tensor data as an example to better illustrate our proposed AMA data formatting: assuming the size of an input graph tensor is $B \times C \times T \times J$, where $B$ is the batch size of mini-batch inference (e.g. $B = 1$ for a single input inference), $C$, $T$ and $J$ are the number of input channels, video frames, and joints, respectively. The first step is to permute the tensor as $J$ columns, with each column of size $C \times B \times T$. Then each column is flattened to $C$ 1D vectors, with each vector of size $B \times T$. For each of such 1D vectors, zeros are padded to make its length equal to the smallest power-of-two integer greater than $B \times T$. The $C$ zero-padded 1D vectors are concatenated and then further stacked to a plaintext vector via a channel-interleaving manner until the space of a plaintext vector can be fully exploited, e.g. the length reaches half of the polynomial degree $N$. Finally, we encrypt such a plaintext as a fully-packed ciphertext $ct_k$, $k \in J$. The detailed process is presented in Algorithm 1 and Fig. 2(a).

**AMA Data Format-Aided Matrix-Matrix Multiplication.** Once the AMA data formatted ciphertext is created, the next step would be to apply it to simplify and speed up the matrix-matrix multiplication introduced by adjacency matrix and convolution operations. Recall Equation 1 in Section 2, a typical ST-GCN layer's computation involves three consecutive *PMult(ct , pt)*: Spatial convolution ($1 \times 1$ plaintext kernel), matrix-matrix multiplication with normalized adjacency matrix $J \times J$ in plaintext, and the Temporal convolution along the dimension $T$ ($K \times 1$ plaintext kernel). Considering the property of $1 \times 1$ Spatial convolutional kernel, we can easily merge this into the adjacency matrix and formulate a new plaintext matrix to reduce one multiplicative depth of $PMult$. Then we can perform the matrix-matrix multiplication based on the new $J \times J$ plaintext matrix $A$. Note that each graph tensor input channel could contain one such matrix.

For row-major formatted ciphertext, as Fig. 2(c) shows (bottom), even with state-of-the-art diagonal encoding method [14], such matrix-matrix multiplication would still involve $2J - 1$ ciphertext rotations (*Rot*). Since the final outcome is the sum of each rotated ciphertext (*Rot(ct,k)*) multiplied by the corresponding diagonal encoded vector ($Pt_k$) from the plaintext matrix $A$, it also brings extra *PMult* and *Add*. In contrast to the row-major data format, our AMA data format can significantly reduce the amount of these HE operations. As Fig. 2(c) (top) demonstrates, first, we decompose the $J \times J$ plaintext matrix $A$ into a series of patterned sparse matrices $A_i$–each $A_i$ contains at most one valid element in each column, and $A = \sum_{i=1}^{m} A_i, m \leq J$. Second, we simply multiply the column-wise fully-packed ciphertext (due to the AMA data formatting) with the valid element of the corresponding column in $A_i$, and then sum the $m$ intermediate column-wise ciphertext to obtain a final ciphertext:

$$ct'_k = \sum_{i=1}^{m} ct_k A_i = \sum_{i=1}^{m} \sum_{k=1}^{J} PMult(ct_{i_k}, a_{i_k k}) \tag{2}$$

Where $a_{i_k k}$ represents the single valid element in column $k$ of $A_i$. Since the process does not require any *Rot*, except the simple column-wise *PMult* with a single plaintext value and final summation, the HE operations can be greatly decreased compared with the row-major format. To be specific, as the example in Fig. 2(c) shows, the sparse adjacency $4 \times 4$ matrix $A$ in a GCN consists of 8 non-zero elements ($a_{11}, a_{13}, a_{14}, a_{23}, a_{32}, a_{41}, a_{42}, a_{44}$) and 8 zeros, while the dense encrypted feature matrix that needs to be multiplied with $A$ has numbers from 1 to 16. For AMA format, as shown in the top part of Fig. 2(c), due to the sparsity of $A$, $A$ can be easily decomposed into two submatrices $A_1 + A_2$ whose column only contains a single non-zero value (e.g. $a_{11}$ in column 1 of $A_1$, $a_{41}$ in column 1 of $A_2$). Then the output ciphertext can be quickly calculated by simply multiplying a constant value in a column of $A_i (i = 1, 2)$ with the corresponding column-wise packed ciphertext in the dense feature matrix, and then sum such column-wise multiplied results from $A_1$ and $A_2$. In this way, the rotation is eliminated. The sparsity of $A$ determines how many submatrices need to be decomposed, and how many $PMults$ are needed. The sparser (e.g. $a_{11}, a_{13}$ become zero) $A$ is, then the less number of $PMult(ct_{i_k}, a_{i_k k})$ is. Thus our AMA format takes advantage of $A's$ sparsity in practical GCN applications to reduce HOCs. In contrast, the row-major format presented in the bottom part of Fig. 2(c), cannot utilize this sparsity for computation overhead reduction due to using a diagonal encoding method to form multiple plaintexts to be multiplied with a corresponding rotated ciphertext.

**Theoretical Analysis of HE Operation Reduction.** We analytically compare the number of HE operations needed for matrix-matrix multiplication between our AMA and row-major data formats. To ensure the evaluation generality, our analysis is conducted by assuming a mini-batch inference

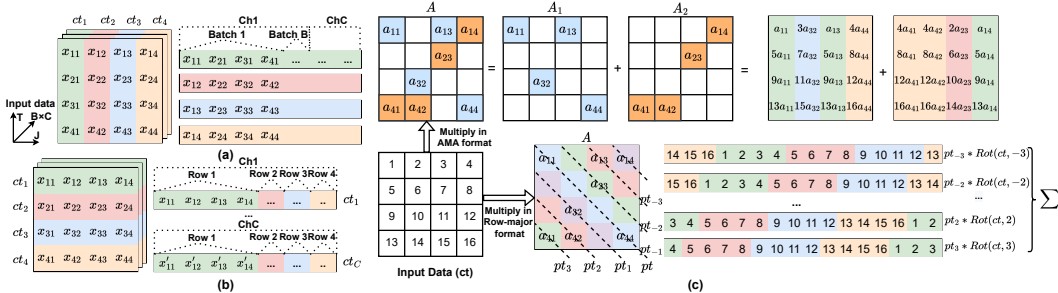

Figure 2: The comparison of data formatting: (a) AMA data format; (b) Row-major format; (c) Matrix-matrix multiplication comparison using AMA and row-major formatted ciphertext.

of multi-channel graph tensor input ($B \times C \times T \times J$) on a typical ST-GCN layer which consists of $C$ input channels and $C$ output channels with $1 \times 1$ convolution kernels. The $J \times J$ matrix $A$ in Eq. 2 can be decomposed into $J$ sub-matrices if $A$ is dense in the worst case. For a fair comparison, we assume the space of a ciphertext–$T \times J$ is fully exploited in both methods. As shown in Fig. 2 (a) and (b), a ciphertext in row-major and AMA format contains one channel data of size $T \times J$, and multiple-channel multiple-batch data of size $B \times T \times c$ ($c = J/B, c \in \mathbb{N}^+$), respectively. This results in the same total amount of ciphertext–$B \times C$ in both methods. The implementations of an example matrix-matrix multiplication using the two methods are presented in Fig. 2 (c). Since AMA formatting increases processing parallelism in SIMD by packing multiple-channel and multi-batch data into one ciphertext, it involves the rotation of the outcomes of matrix-matrix multiplications from different channels and then summing up them to obtain the result for an output channel (Fig. 2 (b)). A ciphertext including data from more channels (a larger $C$) requires more rotations. To further reduce the rotation overhead, we leverage the multi-channel convolutional technique and baby-step strategy from HEAR [22][1]. We also apply them to row-major formatting in evaluation 5. However, the improvement is limited as a row-major formatted ciphertext can only contain single-channel data.

Table 1: Analytical comparison of HE Operation # (AMA v.s. Row-major)

| HE Operation | Description | Total | Complexity |
|---|---|---|---|
| Rot | Row-major | $B \times C \times (2J - 2)$ | $O(B \cdot C \cdot 2J)$ |
| Rot | AMA | $B \times C \times (J/B - 1)$ | $O(C \cdot (J - B))$ |
| PMult | Row-major | $B \times C \times (2J - 1) \times C$ | $O(B \cdot C^2 \cdot 2J)$ |
| PMult | AMA | $J \times J \times (BC/J) \times C$ | $O(B \cdot C^2 \cdot J)$ |
| Add | Row-major | $B \times C \times (2J - 1) \times C - BC$ | $O(B \cdot C^2 \cdot 2J)$ |
| Add | AMA | $J \times J \times (BC/J) \times C - BC$ | $O(B \cdot C^2 \cdot J)$ |

Table 1 reports the detailed breakdown of HE operation numbers for each method (see Appendix A.2 for detailed proof). Overall, AMA data format requires almost half of *PMult* and *Add* operations of that of row-major format, since row-major format needs to compute $2J - 1$ versions of a ciphertext due to rotations in the matrix-matrix multiplication. The number of rotations increases proportionally as the number of channels $C$ that can be included in a ciphertext increase for both methods. However, it is much less than that of row-major. The rotation number difference between AMA and row-major format can be further enlarged when only the batch size $B$ increases. This is because the AMA formatted ciphertext contains data from more batches, resulting in the reduction of rotation times ($JC - BC$). In contrast, the rotation amount of the row-major format grows as $B$ increases. This means that our AMA data formatting performs better when the batch size increases. Moreover, since $A$ is often a sparse matrix in practical ST-GCNs, the number of decomposed matrices $m$ can be much smaller than $J$ for matrix-matrix multiplication, indicating better efficiency than Table 1 which is obtained from a dense matrix. We further validate these observations in Section 5.

## 3.2 Activation Pruning

Applying the Leveled Homomorphic Encryption scheme like CKKS for private inference of a deep network is challenging because of the limited modulus bits for the encrypted ciphertext. This

---

[1]We optimize the $K \times 1$ Temporal convolution following the matrix-matrix multiplication in a similar manner.

means that one LHE encrypted ciphertext has a limited rescale level. For computation like batch normalization, it could be easily absorbed in the adjacent linear computation like convolution and matrix multiplication [13] since batch normalization [17] is a fixed-parameter based linear transformation in inference. However, the nonlinear ReLU activation computation must be replaced by polynomial approximate activation (PAA). However, even a simple degree-2 PAA would consume 2 rescale levels. Given that each GCN layer can contain one or more ReLU activation for accuracy purposes, more rescale levels would be needed for deep GCNs. This increases the polynomial degree to achieve a certain security level, leading to higher computation overhead and inference latency.

| **Algorithm 1** AMA Formatting for Encryption | **Algorithm 2** Activation Pruning |
|---|---|
| 1: **Input:** $X_k = (x_1, x_2, \ldots, x_C) \in \mathbb{R}^{C \times B \times T}$ 
 2: **Output:** $ct_k$, a fully packed ciphertext 

 3: **For** i **from** 1 **to** C 
 4:     $v_i = $ *Flatten* $x_i$ 
 5:     $Pad(v_i) = $ *Padding* $v_i$ *to nearest power of 2* 
 6: **end for** 
 7: $V_k = (Pad(v_1), Pad(v_2), \ldots, Pad(v_C))$ 
 8: $V_{k_{full}} = $ *stack copies of* $V_k$ *to size* $N/2$ 
 9:   **return** $ct_k = Encrypt(V_{k_{full}})$ | 1: **Input: NN Model with** $M$ **activation layers** 
 2: **Output: Optimized Architecture** 
 3: **For** $i$ **from** 1 **to** M 
 4:     $acc_i$=**Train the Model with** $i_{th}$ **activation layer pruned** 
 5: $rank_{act} = [acc_i]$ 
 6: **SORT** $rank_{act}$ **from high to low** 
 7: **For** $i$ **from** 1 **to** $M$ 
 8:     **Fine-tune the network** $Arch_i$ **without top i activations in** $rank_{act}$ 
 9:     *AP-performance* $= (Acc_i, Level_i)$ 
 10: **return** $Arch_i$ *with best AP-performance* |

To achieve the trade-off between inference latency, security level, and accuracy, we investigated the state-of-the-art solution–fine-grained channel-wise PAA for ReLU in SAFENet [28], which is to replace the ReLU activation in the same layer with different polynomial degrees. However, the method has a major limitation when applied to LHE-based private inference. Given a ciphertext could contain data from multiple channels in our AMA data formatting, a mask vector that chooses different PAAs must be utilized to process a ciphertext. As a result, this would consume an extra rescale level and offset the benefit of reduction from the multiplicative level in ReLU. Therefore, we propose to identify and prune ReLU activations based on the fact that removing ReLU activation of some layers in deep networks leads to marginal accuracy loss [18]. For our Leveled HE schemes CKKS, computation overhead from activation is not linearly changed with activation counts different from some existing approaches like CryptoNAS [12] and Delphi [29]. Thus, our target is not to prune as many activations as possible. Instead, we need to take the total multiplication depth, model accuracy, latency, and security level all into consideration in our optimization process.

We name such a technique Activation Pruning (AP). As Algorithm 2 shows, we first replace all the ReLU activations with a 2-degree polynomial activation $ax^2 + bx + c$, in which $(a, b, c)$ is updated during the training process, and train this model's accuracy to a desired level as the baseline. Second, we rank the activation layers according to the accuracy, search the architecture by increasing the number of pruned activation layers and fine-tune the new model to recover the accuracy. A model can be selected based on the accuracy and number of needed rescale levels to trade off between security level and latency.

## 4   Experiment Methodology

**HE parameter setting.** We have two sets of encryption parameters for the experiments without Activation Pruning and with Activation Pruning. For both settings, we choose the scaling factor $\Delta = 2^{33}$ to maintain the inference accuracy. For each level, it will consume 33 bits of ciphertext modulus $Q$. For experiments without AP setting, it requires 21 levels for the whole network architecture. Thus, we set $Q = 740$, and the polynomial degree $N$ should be set to $2^{15}$ to make the security level $\geq 80$ bits. For experiments with the AP setting, the total level is 19 or 17. Therefore, we set $Q = 680$ *or* 600 and the polynomial degree $N = 2^{14}$ to achieve at least 80-bit security level.

**Dataset. NTU-RGB+D** [34] is the current largest dataset with 3D joint annotations for human action recognition task. It contains 56,880 action clips in 60 action classes. The annotations contain the 3D information $(X, Y, Z)$ of 25 joints for each subject in the skeleton sequences. We choose one benchmark **NTU-cross-View** (NTU-XView) as the dataset for our evaluation because this benchmark

Table 2: Model Architecture

| Model | Layer | ST-GCN 1 | ST-GCN 2 | ST-GCN 3 |
|---|---|---|---|---|
| 64-STGCN-3 | Output featuremap size | (256,25) | (128,25) | (128,25) |
| | Channels | 64 | 128 | 128 |
| 128-STGCN-3 | Output featuremap size | (256,25) | (128,25) | (128,25) |
| | Channels | 128 | 256 | 256 |

is a representative human skeleton joints dataset. It contains 37,920 and 18,960 clips for training and evaluation, respectively. For better evaluation, we use 256 frames from the video clip as our input data. Thus, the input tensor with a size of $2 \times 3 \times 256 \times 25$ contains the 25 skeleton-joint information $(X, Y, Z)$ of 2 persons in a video has 256 frames.

**Network Architecture.** ST-GCN [38] is the state-of-the-art GCN architecture for human action recognition. It combines the GCN and CNN to better extract the Spatial and Temporal information than the previous models. In our experiment, we use a stack of 3 ST-GCN layers with a global average pooling layer and a fully-connected layer and study one small net and one large net: 64-STGCN-3 and 128-STGCN-3, where the first number stands for the channel number of the first ST-GCN layer. Table 2 summarizes the network architecture. One ST-GCN layer is composed of one Spatial Conv layer and one Temporal Conv layer. Following the stacked 3 ST-GCN layers are a global average pooling layer and a fully-connected layer. We use Stochastic Gradient Descent (SGD) optimizer with a mini-batch size of 64, a momentum of 0.9, and a weight decay of $e^{-4}$ to train the model for 200 epochs. The initial learning rate is set to 0.01 with a decay factor 0.1.

**Evaluation Setup.** Our experiments are conducted on a machine with AMD Ryzen Threadripper PRO 3975WX with a single thread setting. We use Microsoft SEAL version 3.7.2 [33] to implement a RNS-variant of CKKS [6] scheme. To perform the Temporal convolution, we leverage the baby-step strategy [22] to optimize the multi-channel convolution. The Global Avg Pooling layer and Fully-Connected layer have a small impact on the total latency so we just apply a same implementation with HEAR [22] to compute these two layers.

## 5 Results and Analysis

### 5.1 Activation Pruning Ablation Study

We first analyze the effect of our Activation Pruning (AP) technique in our framework through an ablation study. To evaluate the performance, we optimize our neural network architecture by pruning the activation layers. After the algorithm optimization, we have two types of variants, i.e., 1 AP and 2 AP, where the numbers denote how many activation layers have been pruned. Then, we compare the performance of these optimized architectures on the model inference accuracy and HE inference latency. The results are shown in Table 3.

Despite the architectures having different widths (channel numbers), the original unoptimized architectures (w/o AP) have the best accuracy but also the highest latency. To deal with the multiplicative depth of the unoptimized architectures, we have to set the polynomial degree in the encryption parameter to $2^{15}$, resulting in a tremendous latency increase. By applying AP, the 1 AP (one activation layer pruned) architecture only has a small accuracy loss (1.1-1.5%), while the latency is improved by about $2.3\times$ through saving two levels. This is because reducing two levels allows us to trade off the polynomial degree with a security level decrease and achieve a sweet point. More specifically, we can use $2^{14}$ as our polynomial degree for the optimized depths with an 80-bit (instead of the original 128-bit) security level. If we try to prune one more activation layer (2 AP), the accuracy loss (3.95-4.04%) is larger than the 1 AP variant. Besides, the latency speedup is limited because we can not reduce the polynomial degree further to have at least an 80-bit security level.

### 5.2 AMA Format Effectiveness

To evaluate the effectiveness of the AMA format method, we compare its performance against the row-major formatting with the same encryption parameter setting. As the AP improves the latency for every HE operation, we evaluate the performance of two data formatting methods on AP-optimized

Table 3: Ablation study for AP (Model Architecture Tradeoff) (AMA format)

| Model | | 64-STGCN-3 | 128-STGCN-3 |
|---|---|---|---|
| w/o AP | ACC | 74.25% | 75.31% |
| | Latency | 4273.89s | 10580.41s |
| w/ AP (1 AP) | ACC | 73.12% | 73.78% |
| | Latency | **1863.95s** | **4850.93s** |
| w/ AP (2 AP) | ACC | 70.21% | 71.36% |
| | Latency | 1856.36s | 4831.93s |

Table 4: Ablation study of AMA format with batch size=1

| Model | | 64-STGCN-3 | 128-STGCN-3 |
|---|---|---|---|
| Row-major format | Latency | 2962.46s | 9589.59s |
| | # of Ct | 128 | 256 |
| AMA | Latency | **1863.96s** | **4850.93s** |
| | # of Ct | 100 | 200 |
| Speedup | | 1.59× | 1.97× |

architecture. Then, we compare our AMA format with the row-major format in different batch size settings.

### 5.2.1 Compare with row-major format

As reported in Table 4, our AMA format improves the inference latency by 1.59× for 64-STGCN-3 architectures and 1.97× for 128-STGCN-3 architecture, respectively. Table 5 is a breakdown of HE operations for 64-STGCN-3. From the table, we observe that the AMA format consumes only 31.3% of PMult and Add, comparing with the row-major format. The theoretical analysis of HOC is presented in Table 8 of Appendix A.1.

Table 5: Breakdown for HE operations in different layer of 64-STGCN-3

| Layer | Row-major format | | | | AMA | | | |
|---|---|---|---|---|---|---|---|---|
| | Rot | PMult | CMult | Add | Rot | PMult | CMult | Add |
| Spatial Conv | 6.9K | 623K | - | 622K | 6K | 56K | - | 55K |
| Temporal Conv | 4K | 295K | - | 294K | 7.7K | 230K | - | 230K |
| GlobalAvgPooling | 832 | - | - | 896 | 28 | - | - | 96 |
| FC | 60 | 3.8K | - | 3.8K | 240 | 240 | - | 240 |
| Activation | - | 1.3K | 640 | 640 | - | 1K | 500 | 1K |

Here are two reasons for the HOC reduction. First, the AMA format uses fewer ciphertexts than the row-major format. The row-major format does not fully utilize the ciphertext space as the feature map has a size of $256 \times 25$. In row-major format, the feature map is first converted into a 1D vector with a size of 6400; then zero padding is applied to the right end of this vector to make the size equal to 8192. Therefore, in the resulting encrypted ciphertext, 1792 slots have been wasted. Our AMA format, on the other hand, fully utilizes the ciphertext space, because 256 is a power-of-two number. None of the slots in the ciphertext is wasted. Second, compared to row-major formatting, our AMA format could perform matrix-matrix multiplication with fewer HE operations. For the matrix used in our proposed network, one row-major format ciphertext should perform 19 multiplications for one output channel. However, one AMA format ciphertext only needs to perform 3 multiplications for one output channel. A similar reason holds for the Add operation.

### 5.2.2 Different batch size settings

We analyze our data formatting method in different batch size settings. As described in Table 6, with the batch size increasing, the latency of the row-major format increases linearly as they do not have any parallelism for a mini-batch setting. However, our AMA format allows processing a mini-batch of data in the same ciphertext, which improves the parallelism for a mini-batch setting. Besides, the number of rotations for multi-channel convolution decreases when the batch size increases and the number of other HE operations increases linearly, resulting in a speedup as much as 3.1× for average latency with the batch size growing.

### 5.3 Computation Complexity Evaluation

Table 7 compares CryptoGCN against the state-of-the-art privacy-preserving neural network frameworks (i.e., CHET [11] and Fast-HEAR [22]). Since the previous frameworks and ours are implemented in different environments (different CPUs and number of threads). For a fair comparison, we evaluate the number of the required homomorphic operations for 64-STGCN-3 on the same dataset. In this way, we eliminate the impact of different hardware and software configurations.

Table 6: AMA performance with batch size increase

| Model | Batch size | Row-major | AMA | Average Latency | Speedup |
|---|---|---|---|---|---|
| 64-STGCN-3 | 1 | 2965.46 sec | 1863.95 sec | 1863.95 sec | 1.59 $\times$ |
| 64-STGCN-3 | 2 | 5931.92 sec | 2704.41 sec | 1352.21 sec | 2.19 $\times$ |
| 64-STGCN-3 | 4 | 11852.84 sec | 4390.66 sec | 1097.66 sec | 2.70 $\times$ |
| 64-STGCN-3 | 8 | 23703.68 sec | 7770.91 sec | 971.36 sec | 3.05 $\times$ |
| 64-STGCN-3 | 16 | 45179.33 sec | 14535.23 sec | **908.45 sec** | **3.10** $\times$ |

Table 7: Compare with the previous benchmarks on 64-STGCN-3

| Method | Batch size | HOC | | | | |
|---|---|---|---|---|---|---|
| | | Rot | CMult | PMult | Add | Total |
| CHET | | 16K | 1.28K | 1.3M | 1.29M | 2.61M |
| Fast-HEAR | 1 | 12K | 1K | 923K | 922K | 1.86M |
| **CryptoGCN** | | 14K | 500 | 287K | 287K | 589K |
| CHET | | 32K | 2.56K | 2.6M | 2.59M | 5.23M |
| Fast-HEAR | 2 | 24K | 2K | 1.84M | 1.84M | 3.7M |
| **CryptoGCN** | | **16K** | **1K** | **575K** | **574K** | **1.17M** |

Similar to our work, CHET and Fast-HEAR utilize the ciphertext packing technique to reduce HE computation complexity. They utilize the row-major format as the data representation for the encrypted feature maps. The main difference between Fast-HEAR and CHET is that Fast-HEAR leverages the non-valid space in ciphertext after down-sampling (avg pooling layer) in the CNN model such that Fast-HEAR could have less Homomorphic operation count (HOC).

Neither CHET nor Fast-HEAR is optimized for GCN-based models and the unique matrix multiplication mechanism could significantly increase the HOC. When batch size equals 1, compared to CHET and Fast-HEAR, our AMA formatted ciphertext better utilizes the sparsity of the matrix and significantly reduces the multiplication and addition operations. Specifically, when performing matrix multiplication combined with a multi-channel convolution, the AMA format avoids performing an inner loop for matrix multiplication and hence reduces the amount of PMult and Add operations by 52.5%-66.2%. Furthermore, we pack the graph data into ciphertexts to maximize the use of the slots and prune one activation layer from the original architecture, which reduces the number of CMult by 40%-50%. With the batch size increasing, CryptoGCN reduces 77.4% of the total HOC compared to prior work. A more detailed theoretical comparison of HOC between CrytoGCN, CHET and Fast-HEAR can be found in Table 9 of Appendix A.1.

# 6 Conclusion

Homomorphic encryption (HE) has become an effective way to build privacy-preserving machine learning thanks to the great advancement of HE schemes. In this paper, we build a fast and scalable Leveled HE-based privacy-preserving inference framework optimized for GCN models by leveraging our proposed novel AMA data formatting and model architecture optimization strategy. To the best of our knowledge, this is the first framework supporting private inference for a large skeleton joint data with a 40$\times$ size of previous work–Fast-HEAR using deeper networks. Our solution demonstrates encouraging results for enhancing privacy-preserving inference on GCN models in a cost-effective manner. In the future, we could leverage the multi-threading technique to further improve the encrypted inference latency. We would like to extend the encrypted data to both graph node features and adjacency matrices, as the matrices may also contain a part of sensitive information. By encrypting both graph node features and the associated adjacency matrices, the client's data privacy can be fully guaranteed.

# 7 Acknowledgement

We would like to thank the anonymous reviewers for their constructive comments and suggestions on this work. This work is partially supported by the National Science Foundation (NSF) under Award CCF-2011236, and Award CCF-2006748.

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

# A   Appendix

## A.1   HOC breakdown across layers and other methods

We use the following notations: Number of samples $N$ , Data size $N \cdot C \cdot T \cdot J$, Batch size $B$, Channels on one ciphertext in AMA $U = {}^{R}\!/_{2 \cdot N \cdot T \cdot B}$ , Number of ciphertexts in AMA $N_a = {}^{J \cdot C}\!/_{U}$, Number of ciphertexts in row-major format $N_r = C \cdot N \cdot B$ , Output channel $O$ , polynomial degree $R$, Spatial-conv layers $S_p$, Temporal-conv layers $T_e$, Activation layers $A$, Kernel size $K$, Valid matrix elements number $V$, Diagonal decomposition number $D$, Number of output classes $C_s$. The HOC comparison across layers between AMA format and row-major format is presented in Table 8. The model-level HOC comparison with other SOTA methods is listed in Table 9.

Table 8: HOC layer breakdown for AMA format

| Layer | Type | Rot | PMult | CMult | Add |
|---|---|---|---|---|---|
| S-Conv | | $J \cdot (S_p + 1) \cdot (O/U) \cdot (U-1)$ | $N_a \cdot (V/J) \cdot O \cdot S_p$ | 0 | $(N_a \cdot (V/J) \cdot O - N_a) \cdot S_p$ |
| T-Conv | | $(N_a \cdot (K-1) \cdot (T_e + 1) + J \cdot (U-1) \cdot (O/U) \cdot S_p$ | $N_a \cdot O \cdot K \cdot (T_e + 1)$ | 0 | $(N_a \cdot O \cdot K - N_a) \cdot (T_e + 1)$ |
| GAP | AMA | $C/U \cdot log(T/2)$ | 0 | 0 | $C/U \cdot (J-1)$ |
| FC | | $C/U \cdot C_s$ | $C/U \cdot C_s$ | 0 | $(C/U) \cdot C_s$ |
| Activation | | 0 | $2 \cdot N_a \cdot A$ | $N_a \cdot A$ | $2 \cdot N_a \cdot A$ |
| S-Conv | | $N_r \cdot (D-1) \cdot S_p$ | $N_r \cdot D \cdot C \cdot (S_p + 1)$ | 0 | $(N_r \cdot O \cdot K - N_r) \cdot (S_p + 1)$ |
| T-Conv | | $N_r \cdot (K-1) \cdot (T_e + 1)$ | $N_r \cdot K \cdot O \cdot (T_e + 1)$ | 0 | $(N_r \cdot K \cdot O - N_r) \cdot (T_e + 1)$ |
| GAP | Row-major | $N_r/N \cdot log(R/2)$ | 0 | 0 | $N_r/N + N_r/N \cdot log(R/2)$ |
| FC | | $C_s$ | $N_r/N \cdot C_s$ | 0 | $N_r/N \cdot C_s$ |
| Activation | | 0 | $N_r \cdot 2 \cdot A$ | $N_r \cdot A$ | $N_r \cdot A$ |

Table 9: HOC Comparsion with other methods

| Methods | Rot |
|---|---|
| CHET | $N_r \cdot (D-1) \cdot (S_p + 1) + N_r \cdot (K-1) \cdot (T_e + 2) + N_r \cdot log(R/2) + C_s$ |
| F-HEAR | $N_r \cdot (D-1) \cdot S_p + N_r \cdot (K-1) \cdot (T_e + 1) + N_r/N \cdot log(R/2) + C_s$ |
| CryptoGCN | $J \cdot (S_p + 1 + T_e) \cdot (O/U) \cdot (U-1) + (N_a \cdot (K-1) \cdot (T_e + 1) + C/U \cdot log(T/2) + C/U \cdot C_s$ |

| Methods | PMult |
|---|---|
| CHET | $N_r \cdot D \cdot O \cdot (S_p + 2) + N_r \cdot K \cdot O \cdot (T_e + 4) + N_r \cdot 2 \cdot A \cdot 2 + N_r/N \cdot C_s$ |
| F-HEAR | $N_r \cdot D \cdot O \cdot (S_p + 1) + N_r \cdot K \cdot O \cdot (T_e + 1) + N_r \cdot 2 \cdot (A + 3) + N_r/N \cdot C_s$ |
| CryptoGCN | $N_a \cdot (V/J) \cdot O \cdot S_p + N_a \cdot O \cdot K \cdot (T_e + 1) + C/U \cdot C_s + 2 \cdot N_a \cdot A$ |

| Method | Add |
|---|---|
| CHET | $(N_r \cdot D \cdot O - N_r) \cdot (S_p + 2) + (N_r \cdot K \cdot O - N_r) \cdot (T_e + 4) + N_r \cdot A \cdot 2 + N_r/N + N_r/2 \cdot log(R/2) + N_r/N \cdot C_s$ |
| F-HEAR | $(N_r \cdot O \cdot K - N_r) \cdot (S_p + 1) + (N_r \cdot K \cdot O - N_r) \cdot (T_e + 1) + N_r \cdot (A + 3) + N_r/N + N_r/N \cdot log(R/2) + C_s \cdot N_r/N$ |
| CryptoGCN | $(N_a \cdot (V/J) \cdot O - N_a) \cdot S_p + (N_a \cdot O \cdot K - N_a) \cdot (T_e + 1) + C/U \cdot (J-1) + (C/U) \cdot C_s + 2 \cdot N_a \cdot A$ |

| Method | CMult |
|---|---|
| CHET | $N_r \cdot A \cdot 2$ |
| F-HEAR | $N_r \cdot (A + 3)$ |
| CryptoGCN | $N_a \cdot A$ |

## A.2   Theoretical Comparison of HOC for Matrix-Matrix Multiplication

We explain the theoretical results in Table 1 by following the same assumption in Section 3.1.

**Row-major format:**

For one ciphertext, it represents the data from one channel. In order to multiply with a $J \times J$ dense matrix, we need to rotate each ciphertext by $2J - 2$ times. To obtain $C$ output channel data, all existing $BC$ ciphertexts need to perform $2J - 2$ times multiplications, then we get $BC$ resulted ciphertexts by summing up all the ciphertexts. The computation overhead for the three HE operations can be expressed in Equation 3:

$$\begin{aligned} Rotation &= B \cdot C \cdot (2J - 2) \\ PMult &= C \cdot B \cdot C \cdot (2J - 2) \\ Add &= C \cdot B \cdot C \cdot (2J - 2) - B \cdot C \end{aligned} \tag{3}$$

**AMA format:**

The rotation operation is only used to sum up all the input channel data, because the ciphertext with AMA format contains $J/B$ channel data. For each ciphertext, it needs to rotate $(J/B - 1)$ times. To get $C$ output channel data, each ciphertext needs to perform $J$ times $PMult$. Then we get $BC$ resulted ciphertexts by summing up all the ciphertexts. The computation overhead for three HE operations can be found in Equation 4:

$$\begin{aligned} Rotation &= B \cdot C \cdot ({}^{J}\!/_{B} - 1) \\ PMult &= B \cdot C \cdot J \cdot C \\ Add &= B \cdot C \cdot J \cdot C - B \cdot C \end{aligned} \tag{4}$$

