# OpenReview forum: "CryptoGCN: Fast and Scalable Homomorphically Encrypted Graph Convolutional Network Inference"
_NeurIPS.cc/2022/Conference — NeurIPS 2022 Accept_

### Official Review · Reviewer_2m5X · 2022-07-11

**Rating:** 7
**Confidence:** 3
**Soundness:** 3 good
**Presentation:** 2 fair
**Contribution:** 3 good

**Summary:**

This work deals with two core techniques to efficiently perform GCN networks on HE with a reduced amount of homomorphic operations. First, since the GCN networks usually use the multiplication with the sparse plaintext matrix, the authors devise AMA data formatting to reduce the homomorphic operations using this sparsity of the matrices. Second, they pruned some activation functions less affecting the performance of the GCN networks, and also proposed a method to find these less-affecting activation functions. With these techniques, the authors successfully reduced the number of homomorphic operations compared to the prior works applied to GCN networks, where these works were not optimized for the GCN networks.

**Questions:**

- Will the implementation of this model be open to the public after acceptance?

**Limitations:**

There is no negative societal impact.

**Strengths And Weaknesses:**

Strengths
- This work deals with the GCN network on HE for the first time. It is desirable to optimize the important networks on HE other than traditional CNN models.
- The authors observe the sparsity of the matrix in the GCN network and effectively devise the packing method using this sparsity of the matrix. The academic contribution of this technique is good enough.
- The activation pruning technique is not limited to the GCN network and can be applied to more general frameworks, and this technique can be a useful practical technique for reducing the number of homomorphic operations.
- The numerical comparison and theoretical analysis are well shown. Even though this work proposed the first GCN network on HE, they compared the proposed model with the prior work applied to GCN networks. Thus, these techniques were well proven to be effective in GCN networks on HE compared to the prior work for traditional CNN networks.

Weaknesses
- The AMA data formatting is not sufficiently presented. Algorithm 1 is too compact and abstract to be understood exactly, and there is no algorithm for AMA data format-aided matrix-matrix multiplication. Although the authors explain these algorithms in Figure 2, the figures usually play the role of only supplementary material to ease the understanding of the algorithm. I know there is no sufficient margin for this algorithm, but the author should add the specific and detailed pseudocode for these algorithms in Appendix for readers who want to understand the algorithms exactly.

---

> ### Author Response · Authors · 2022-08-02
> **Response**
>
> Thanks for reviewing our paper and giving constructive feedback for the presentation issues!
>
> Due to limitation of pages, we try to use the formula (2) in our draft to represent the ciphertext multiplication with the matrix decomposition plaintext and then get the result. We would like to keep updating the algorithm and looking into more discussions.
>
> We would like to take your suggestions and improve the algorithm 1 presentation with more detailed notations about the actual matrix multiplication part in the Appendix. That would make this algorithm more understandable.  We will revise the paper for final submission if accepted. As this area is emerging and full of practical value in application, we would like to open source our code after acceptance and wish to make a contribution to the whole community.

---

### Official Review · Reviewer_YLFb · 2022-07-12

**Rating:** 6
**Confidence:** 4
**Soundness:** 3 good
**Presentation:** 2 fair
**Contribution:** 2 fair

**Summary:**

The paper studies private inference of graph convolutional neural networks using leveled homomorphic encryption. The main contributions are, i) a data encoding scheme that is aware of the adjacency matrix of the graph and allows for more efficient spatial convolution that exploits the graph structure to perform matrix-matrix multiplication, and ii) pruning ReLUs to reduce computation overhead.

The method is evaluated on the NTU-XVIEW skeleton joint dataset and demonstrates lower latency compared to existing baseline methods.



**Questions:**


- In Fig 2c, is the 4x4 matrix with numbers from 1-16 the adjacency matrix? Why is it dense?
- Can the computational complexity in Tables 5 and 7 (i.e., rotations, PMult, CMult, etc.) be provided in terms of the number of samples $n$, data dimension $d$, batch-size $b$, etc. instead of just showing the numerical values?
- The abstract says, "we develop an approach that can effectively take advantage of the sparsity of matrix operations in GCN inference to significantly reduce the computational overhead". It is not clear to this reviewer where exactly is the sparsity of $A$ being leveraged.
- When you compare CryptoGCN with CHET and Fast-HEAR, you report the computation complexity, i.e. the number of homomorphic operations. I am just wondering if CryptoGCN’s operations take the same time as CHET and Fast-HEAR? Although CHET and Fast-HEAR are both using CKKS, do you maintain the same cryptographic parameters? Can the row-major format use a smaller ring size and be efficient as well?
- This is more of a suggestion. Algorithm 2 is heuristic and not efficient. You train the model with $i$-th activation layer pruned
one by one and sort the performance. Activation pruning is a combination problem. If you only estimate the importance of each ReLU, it is not accurate. Both SafeNet and Delphi use PBT to solve this problem. PBT should be more efficient and can get better solutions than the proposed Activation Prune algorithm.

Will revise the rating of the paper based on the author's rebuttal.

**Limitations:**

- The paper does not discuss the limitations of the current work nor does it discuss any negative societal impact. I guess since the paper is focused on private inference, there is limited negative societal impact beyond making "black-box" GCNs even more "black-box" via encryption.

**Strengths And Weaknesses:**

**Strengths:**
+ Important and well-motivated problem.
+ A customized data formatting of GCN over encrypted data under the CKKS scheme. Because CKKS supports SIMD and it is necessary to find a customized way to encode data that is fed into a GCN so that it can better take advantage of SIMD. Previous work majorly focuses on CNNs. There are some presentation issues though as described below.
+ Experiments show a clear improvement in latency compared to the baselines.

**Weaknesses:**
- Pruning ReLU is not a new idea. Existing methods such as SafeNet (which searches channel-wise mixed-degree approximation of ReLU including pruning), CryptoNAS (explicitly focuses on removing ReLU layers), and Delphi also shares similar ideas and performs layer-wise pruning.
- This is more of a presentation issue rather than a fatal flaw. The AMA-based matrix-matrix multiplication shown in Fig 2(c) is not clear to this reviewer. The description in lines 178-180 is not clear in terms of how exactly is this being computed. An algorithm was provided for the AMA data formatting but not for the matrix-matrix multiplication. To summarize, while the AMA data formating is very clear, how it is being used for matrix-matrix multiplication is not clear from the description in the paper.

---

> ### Author Response · Authors · 2022-08-02
> **Response**
>
> Thanks for reviewing our paper and giving a lot of meaningful suggestions and comments. We agree that ReLU pruning is not a new idea. Our major contribution here is (see Algorithm 2 on page 6) to explore the trade-off among model accuracy, multiplication level, and security level in the context of HE-based GCN inference by ReLU pruning. Note that, for HE schemes like CKKS, computation overhead from activation is not linearly changed with activation counts different from some existing approaches like CryptoNAS and Delphi. Our target is not to prune as many activations as possible. Instead, we need to take the total multiplication depth, model accuracy, latency, and security level all into considerations in our optimization process. For SafeNet, the channel-wise mixed-degree polynomial approximation for the ReLU function may not work well for CKKS encrypted ciphertext because one ciphertext usually packs and encrypts multi-channels data. For example, if one channel uses a 3-degree approximation, and others use a 2-degree approximation, the ciphertext needs to do cube computation for the whole ciphertext due to this 3-degree approximation in one channel. However, this is unnecessary for other channels' data, and therefore causes an additional multiplication depth for the whole ciphertext.
>
> Q1:
>
> This is a representation issue. We tried to explain the difference between the row-major format and our AMA-format ciphertexts performing matrix multiplication in Fig 2 (c). This 4x4 matrix with numbers 1 to 16, represents the encrypted data which need to multiply with the adjacency matrix (with elements a11, a13, a14…and zeros). Thus, it is a dense matrix. We will update Fig. 3 to avoid the confusion.
>
> Q2:
>
> We use the following notations: Number of samples $N$, Data size $N·C·T·J$, Channels on one ciphertext in AMA  $U=R/(2·N·T·B) $, Batch size $B$, Number of ciphertexts in AMA  $Na=J·C/U$, Number of ciphertexts in row-major format $Nr=C·N·B$, Output channel $O$, Ring size $R$, Spatial-conv layers $Sp$, Temporal-conv layers $Te$, Activation layers $A$, Kernel size $K$, Valid matrix elements number $V$, Diagonal decomposition number $D$, Number of output classes $Cs$
>
> Table 5
>
> | Layer in AMA format |  Rot |  PMult |  CMult |  Add |
> |---|---|---|---|---|
> | spatial-conv | J·(Sp+1)·(O/U)·(U-1) | Na·(V/J)·O·Sp | 0 | (Na·(V/J)·O-Na)·Sp |
> | temporal-conv | (Na·(K-1)·(Te+1)+ J·(U-1)·(O/U)·Sp | Na·O·K·(Te+1) | 0 | (Na·O·K-Na)·(Te+1) |
> | pooling | C/U·log(T/2) | 0 | 0 | C/U·(J-1) |
> | fc | C/U·Cs | C/U·Cs | 0 | (C/U)·Cs |
> | activation | 0 | 2·Na·A | Na· A | 2· Na·A |
>
> | Layer in row-major |  Rot |  PMult |  CMult |  Add |
> |---|---|---|---|---|
> | spatial-conv | Nr·(D-1)·Sp | Nr·D·C·(Sp+1) | 0 | (Nr·O·K-Nr)·(Sp+1) |
> | temporal-conv | Nr·(K-1)·(Te+1) | Nr·K·O·(Te+1) | 0 | (Nr·K·O-Nr)·(Te+1) |
> | pooling | Nr/N·log(R/2) | 0 | 0 | Nr/N+Nr/N·log(R/2) |
> | fc | Cs | Nr/N·Cs | 0 | Nr/N·Cs |
> | activation | 0 | Nr·2·A | Nr·A | Nr·A |
>
> Table 7
> | Method | Rot | PMult | CMult | Add |
> |---|---|---|---|---|
> | CHET | Nr·(D-1)·(Sp+1)+Nr·(K-1)·(Te+2) +Nr·log(R/2)+CS | Nr·D·O·(Sp+2)+Nr·K·O·(Te+4) +Nr·2·A·2+Nr/N·Cs | Nr·A·2 | (Nr·D·O-Nr)·(Sp+2)+(Nr·K·O-Nr)·(Te+4) +Nr·A·2+Nr/N+Nr/2·log(R/2)+Nr/N·Cs |
> | Fast-HEAR | Nr·(D-1)·Sp+Nr·(K-1)·(Te+1) +Nr/N·log(R/2)+Cs | Nr·D·O·(Sp+1)+Nr·K·O·(Te+1) +Nr·2·(A+3)+Nr/N·Cs | Nr·(A+3) | (Nr·O·K-Nr)·(Sp+1)+(Nr·K·O-Nr)·(Te+1) +Nr·(A+3)+Nr/N+Nr/N·log(R/2)+Cs·Nr/N |
>
> Our method HOC could be got by summed up from the breakdown. We would like to add this part of the result in the revised paper to address the reviewer’s concern.
>
> Q3:
>
> As the example in Fig. 2(c) shows, the sparse adjacency 4x4 matrix A in a GCN consists of 8 non-zero elements (a11, a13, a14, a23, a32, a41, a42, a44) and 8 zeros, while the dense encrypted feature matrix that needs to be multiplied with A has numbers 1 to 16. For AMA-format, as shown in the top part of Fig.2 (c), due to the sparsity of A, A can be easily decomposed into two submatrices A1/A2 whose column only contains a single non-zero value (e.g. a11 in column 1 of A1, a41 in column 1 of A2). Then the output ciphertext can be quickly calculated by simply multiplying a constant value in a column of Ai (i=1,2)  with the corresponding column- wise packed ciphertext in the dense feature matrix, and then sum such column-wise multiplied results from A1 and A2. In this way, the rotation is eliminated. The sparsity of A determines how many submatrices need to be decomposed, and how many PMults are needed. The sparser (e.g.  a11, a13 become zero) A is, then the less numbers of Ai and column-wise PMult, (a11*column ciphertext) will be. Thus our AMA-format takes advantage of A’s sparsity in practical GCN applications to reduce HOCs. In contrast, the row-major format presented in the bottom part of Fig. 2(c), cannot utilize this sparsity for computation overhead reduction due to using a diagonal encoding method to form multiple plaintexts to be multiplied with a corresponding rotated ciphertext.

---

> > ### Author Response · Authors · 2022-08-02
> > **Supplementary Response**
> >
> > Q4:
> >
> > We compare our work with the two baselines under the same cryptographic parameters and the same security level. Each HE operation is assumed to have the same latency under the same running environment.
> > Regarding the security level, since the CKKS ring size is related to other parameters (e.g. primes) as well as the total multiplicative depth. Since the two formatted ciphertexts have the same multiplication depth and same primes, the row-major format ciphertext could not use a smaller size than the AMA-format.
> >
> > Q5:
> >
> > Thanks for your excellent suggestion. As shown in our experimental results (see Table 3 on page 8), Algorithm 2 can achieve over 2.19× improvement on inference latency than the baseline results with approximate 1% accuracy loss. We expect that incorporating the PBT may further improve the performance of this algorithm and it will be interesting to study how much improvement can be achieved using this approach in future work.

---

> > > ### Author Response · Authors · 2022-08-07
> > > **Further Comment**
> > >
> > > Dear reviewer YLFb, thank you for providing the constructive feedback. We have provided our initial responses for you to review. Should you have any further questions regarding to our responses, please kindly let us know. We are open for any further discussion. Thanks for your great efforts!

---

### Official Review · Reviewer_RkGQ · 2022-07-13

**Rating:** 5
**Confidence:** 3
**Soundness:** 2 fair
**Presentation:** 3 good
**Contribution:** 2 fair

**Summary:**

This work studies the privacy-preserving graph neural networks (GNN) using fully homomorphic encryption. Existing works mainly focus on protecting the data privacy of convolutional neural networks (CNN) using fully homomorphic encryption. This paper tries to explore if fully homomorphic encryption can be used to protect the data privacy pf graph neural networks.  The paper argues that directly applying techniques in privacy-preserving CNN will not work since graph neural network operations brings additional matrix multiplication. To reduce the computational overhead, the paper mainly proposed two techniques including Adjacency Matrix-Aware (AMA) multiplication, and activation pruning techniques.

**Questions:**

1. The difference between CNN and GCN is not clear. Although the paper uses figure 1 to highlight that GCN has one more matrix multiplication, the GCN and CNN have the same operators. How to design efficient homomorphic encryption matrix multiplication is not novel. Please try to explain the difference compared to existing works like [2][3][4]. It is important to know why existing techniques in FHE-based CNN or [2][3][4] do not work for the GCN. It is also important to know what is the motivation of the proposed AMA-based matrix multiplication. Is it specific for GCN?

2. The reproduction is not clear to me. It is still not clear why the activation pruning does not significantly hurt the accuracy.

**Limitations:**

There are some overclaims. For example, the author claims “this is the first work that builds the PPML pipeline for GCN-based models”. As I know, there are multiple existing PPML for GCN based networks, like [1].

**Strengths And Weaknesses:**

Strengths:

1. Graph neural networks are ubiquitous now and protecting the data privacy of input data is strongly needed.
2. Fully homomorphic encryption (FHE) is a popular method to enable a privacy-preserving computing on cloud. Combining FHE and GNN is a promising direction.
3. The paper is well-written.

Weakness:
1. There are some overclaims. For example, the author claims “this is the first work that builds the PPML pipeline for GCN-based models”. As I know, there are multiple existing  PPML for GCN based networks, like [1].

[1] Igamberdiev, T., & Habernal, I. (2021). Privacy-preserving graph convolutional networks for text classification. arXiv preprint arXiv:2102. 09604.

2. The difference between CNN and GCN is not clear. Although the paper uses figure 1 to highlight that GCN has one more matrix multiplication, the GCN and CNN have same operators. How to design efficient homomorphic encryption matrix multiplication is not novel. Please try to explain the difference compared to existing works like [2][3][4]. It is important to know why existing techniques in FHE-based CNN or [2][3][4] do not work for the GCN. It is also important to know what is the motivation of the proposed AMA-based matrix multiplication. Is it specific for GCN?

[2]Chiang, J. (2022). A Novel Matrix-Encoding Method for Privacy-Preserving Neural Networks (Inference). arXiv preprint arXiv:2201.12577.

[3] Jiang X, Kim M, Lauter K, Song Y. Secure Outsourced Matrix Computation and Application to Neural Networks. Conf Comput Commun Secur. 2018 Oct;2018:1209-1222. doi: 10.1145/3243734.3243837. PMID: 31404438; PMCID: PMC6689419.

[4] Cui, J., Chen, C., Lyu, L., Yang, C., & Li, W. (2021). Exploiting Data Sparsity in Secure Cross-Platform Social Recommendation. Advances in Neural Information Processing Systems, 34, 10524–10534.


3. Paper checklist. The reproduction is not clear to me. It is still not clear why the activation pruning does not significantly hurt the accuracy.

---

> ### Author Response · Authors · 2022-08-02
> **Response**
>
> Thanks for providing your constructive feedback. In the abstract of our paper, we claim that “we take an initial attempt … and develop CryptoGCN–a homomorphic encryption (HE) based GCN inference framework”.  It is true that privacy protection can be achieved from different schemes such as Differential Privacy (DP) and Homomorphic Encryption (HE). They have different pros and cons, and may be adopted  for different applications. For example, in the reference provided by the reviewer (i.e., reference [1]), the authors use DP to train the model with adding noise to achieve privacy protection for GCN-based NLP applications. In this paper, we use HE, which is a completely different scheme from DP for privacy protection. The major drawback of HE is its high computational overhead, which is exactly what we intend to address in this paper, with a focus on applications based on the Spatial Temporal-GCN (ST-GCN) model.  In our original draft, at the end of the introduction section, we made the claim that “this is the first work that builds the PPML pipeline for GCN-based models with HOC (homomorphic operation count) decrease by… ”. To avoid confusion, we will update it as “this is the first work that builds the HE-based PPML pipeline for ST-GCN models with HOC (homomorphic operation count) decrease by…” in our revised version.
>
> Q1:
>
> One major difference between the deep GCN model and the deep CNN model is that, to incorporate the graphic information in the convolution, each graph convolution layer in the deep GCN model would involve an adjacency matrix multiplication operation. Unfortunately, the HE scheme (such as the CKKS used in this paper) is very sensitive to the multiplication operations. The increased multiplication operations not only demand extra significant computational cost, but also increase the multiplication depths of the total computation circuit, which would further increase the encryption parameters (e.g. increased ring size and primes) to maintain the same security levels.
> One intuitive approach to deal with the adjacency matrix multiplication operation at each graph convolution layer is to employ existing HE matrix multiplication methods (e.g. reference [2, 3] provided by the reviewer). They may work for shallow GCN models, but can be problematic for deep GCN models. With no optimization, these results would surely be inferior to the ones in [5], which we used in our paper as the baseline results for comparison. Instead, we take advantage of the sparsity in the adjacency matrix, which is very common especially for applications based on spatial temporal GCN (ST_GCN) model and can dramatically reduce the number of HE operations.
> To better address the reviewer’s concern, we performed a quick simulation under the same parameter settings and compared the latency results for the approaches mentioned above. The results are shown in Table 1 below.
>
> Table 1
>
> | [2] latency | [3] latency  | [5] latency | Our latency |
> |---|---|---|---|
> | 5815.352s | 7552.949s | 2965.46s | 1863.95s |
>
> In fact, we also found that the approaches in [2, 3] increase the total multiplication depths by 3 & 6 respectively, which led to a higher ring size for encryption to maintain at least 80-bit security level. As shown in Table 1, the latency results from using the HE-matrix multiplication methods in [2,3] are worse than that by [5] and by our approach proposed in this paper. In light of the reviewer’s comments, we can add additional results (e.g. those based on approaches in [2,3] in the revised paper if there is enough space.)
> Note that the approach in reference paper [4] also employs the sparsity location vector to avoid unnecessary computations when performing sparse matrix multiplication. To this end, they developed a PIR protocol to extract the required non-zero elements from the ciphertexts in their multi-party computation (MPC) setting, which is not applicable in our HE environment at all since decryption at the inference stage is not possible.
>
> Q2:
>
> Existing work (such as [6,7]) has clearly shown that different activation layers have different contributions to the model accuracy, and thus pruning some activation layers may not significantly affect the accuracy loss. Based on this observation, we developed the Algorithm 2 (see page 6) with the objective to make the tradeoff among different optimization goals including model accuracy, total multiplication depth and security level. We will also open source our code for reproduction convenience if the paper is accepted.
>
> [5] Shai Halevi and Victor Shoup. Algorithms in helib. In Annual Cryptology Conference, pages 370 554–571. Springer, 2014.
>
> [6] ​​Ghodsi, Zahra, et al. "Cryptonas: Private inference on a relu budget." Advances in Neural Information Processing Systems 33 (2020): 16961-16971.
>
> [7] Jha, Nandan Kumar, et al. "DeepReDuce: Relu reduction for fast private inference." International Conference on Machine Learning. PMLR, 2021.

---

> > ### Author Response · Authors · 2022-08-07
> > **Further Comment**
> >
> > Dear reviewer RkGQ, thank you for providing constructive feedback. We provided our initial response for you to review. Should you have any further questions after reading the response, please kindly let us know. We are happy to discuss it with you. Thanks for your great effort!

---

> > > ### Comment · Reviewer_RkGQ · 2022-08-07
> > > **Increased score**
> > >
> > > Thanks for the authors' response which helps me better understand the paper's contribution. I would recommend the authors incorporate the comparison in the response into the next-version manuscript and explain the reason why the proposed method is more efficient. Also, considering the authors' promise to open their codes, I increased the score.

---

> > > > ### Author Response · Authors · 2022-08-08
> > > > **Thanks to reviewer RkGQ great support to our work**
> > > >
> > > > We sincerely appreciate your great support to our work, and would like to incorporate the comparison and relevant discussion into the revised version as you suggested. In particular, this will be included in Section 5.2 AMA-format effectiveness. Also, we will add a public GitHub link of our source code in the revised paper for the research community to further advance this emerging research field.

---

### Meta-Review · Area_Chair_tQWS · 2022-08-25

**Recommendation:** Accept
**Confidence:** Certain

**Metareview:**

The reviewers appreciate the importance of both GCNs as an application, as well as the application of FHE to make their computations secure. The authors are strongly encouraged to:
1) Include the comparison to the prior work that reviewer RkGQ identified in the camera ready version.
2) Discuss complexity, and include the extended table, at the very least in the supplement.
3) Update Fig. 3, and surface the main takeaways from responses to issues raised by Reviewer YLFb to the main text.


**Award:**

No

---

### Decision · Program_Chairs · 2022-09-14

Accept